# Diversity and DNA Barcode Analysis of Chironomids (Diptera: Chironomidae) from Large Rivers in South Korea

**DOI:** 10.3390/insects13040346

**Published:** 2022-03-31

**Authors:** Hyo Jeong Kang, Min Jeong Baek, Ji Hyoun Kang, Yeon Jae Bae

**Affiliations:** 1Department of Life Science, Graduate School, Korea University, Seoul 02841, Korea; kanghj0413@korea.ac.kr; 2National Institute of Biological Resources, Incheon 22689, Korea; whitechiro@korea.kr; 3Korean Entomological Institute, Korea University, Seoul 02841, Korea; jihyounkang@korea.ac.kr; 4Division of Environmental Science and Ecological Engineering, College of Life Sciences, Korea University, Seoul 02841, Korea

**Keywords:** Chironomidae, distribution, DNA barcode library, *COI*, large river, South Korea

## Abstract

**Simple Summary:**

We aimed to identify chironomid species collected from four large rivers in South Korea, construct a corresponding DNA barcode library, and examine the distribution and community structure of the identified riverine species. Adult chironomids were identified morphologically, and their *COI* nucleotide sequences were used to verify species identification and construct a DNA barcode library. The resulting *COI* library effectively discriminated >90% of riverine Chironomidae in South Korea. The distributional aspects of chironomid species in the four large rivers of South Korea are also discussed.

**Abstract:**

Most large rivers in South Korea run through major cities, which often experience many environmental problems, including outbreaks of non-biting midges (Diptera: Chironomidae). However, chironomid species inhabiting large rivers have not been thoroughly investigated. We aimed to identify chironomid species collected from the four main large rivers in South Korea, construct a corresponding DNA barcode library, and examine the distribution and community structure of the identified riverine species. Adult chironomids were collected from nine sites along the rivers by using sweep nets and light traps during June and August 2015. Adults were morphologically identified, and *COI* nucleotide sequences were generated to verify the species identification and construct a DNA barcode library. The distribution and community structure of the identified species were also analyzed. A total of 124 *COI* sequences were established from 37 species belonging to 19 genera, and the resulting DNA barcode library effectively discriminated >90% of riverine Chironomidae in South Korea. Ten species, which are considered indicator species for large rivers, were collected from all four rivers. In addition, members of the subfamily Chironominae were collected more frequently than members of other subfamilies, with *Tanytarsus tamagotoi* being the most common and widespread chironomid species in South Korea. The DNA barcode library developed in this study will facilitate environmental studies of large rivers, such as biomonitoring chironomid larvae.

## 1. Introduction

The larvae of non-biting midges (Diptera: Chironomidae) are among the most abundant and widely distributed benthic macroinvertebrates in freshwater ecosystems, with 15,000–20,000 species [1,2], which together comprise 50% of the total macroinvertebrate richness and abundance in freshwater ecosystems [3]. Approximately 300 (~1.5%) of these adult species have been reported in the Korean Peninsula [4,5,6,7,8,9,10,11,12,13]. However, the knowledge on larval or pupal stages is minimal, with only a few species being investigated because of the lack of studies on key morphological characters.

Chironomid larvae inhabit the most organically rich sediments of large rivers and are crucial benthic macroinvertebrates for biomonitoring rivers that supply freshwater resources to large cities or metropolises. In South Korea, these rivers include the nation’s four main large rivers: the Han River (Seoul: population 9.6 million), Geum River (Daejeon: 1.5 million), Yeongsan River (Gwangju: 1.4 million), and Nakdong River (Busan: 3.3 million; Daegu: 2.3 million) [14] (Figure 1).

Recent environmental changes in river systems, such as increases in lentic areas due to dam construction, could result in increasingly favorable environments for chironomid larvae that prefer organic-rich fine sediments that are often present in areas with slow currents and in the lentic areas of large rivers [15,16,17]. For example, 16 massive concrete weirs were recently constructed as part of the “Four Major Rivers Project” (construction period 2009–2011) in South Korea, and after this construction, >90% of the benthic macroinvertebrate fauna in the corresponding impoundment areas were replaced, in terms of richness and abundance, by chironomid larvae and oligochaete worms [18]. Additionally, larval chironomids are sometimes found in water purification plants and tap water in Korea [19].

With an increase in chironomid larvae, the emergence rate of adults also increases, causing a nuisance to people. Although adult chironomids do not bite or spread diseases as mosquitoes do, they occur in large numbers in rivers and streams. Public health is negatively affected by debris in the form of chironomid remains in these water bodies that can contribute to respiratory allergies and the breeding and transport of pathogens [20].

Despite the ecological and environmental importance of riverine chironomids, studies on chironomids in Korea have been limited to the taxonomy of specific groups [6,10,21,22] or local fauna [9,11], and intensive investigations of riverine chironomid species have yet to be conducted. To facilitate such investigations, fully evaluating the diversity of riverine chironomids and accurately identifying chironomid larvae during large river biomonitoring in South Korea are necessary, and a chironomid DNA barcode library, based on adult chironomid specimens from South Korea’s main large rivers, is critical.

In this study, we aimed to identify chironomid species collected from the four main large rivers in South Korea, construct a corresponding DNA barcode library, and examine the species distributions and community structures of the identified riverine species. We expect the resulting DNA barcode library to be a useful reverse taxonomy tool for those involved in diverse taxonomic and ecological studies of chironomids in the future.

## 2. Materials and Methods

### 2.1. Sampling Locations and Methods

South Korea’s four main large rivers, namely the Han River (length: 481 km; basin area: 26,018 km^2^), Geum River (length 397 km, basin area 9810 km^2^), Yeongsan River (length 116 km, basin area 2798 km^2^), and Nakdong River (length 521 km, basin area 23,871 km^2^) [23], provide the most freshwater resources to South Korea’s major cities (Figure 1). Chironomid sampling was performed at nine sites along these four major rivers (Figure 2a–d), and all sampling sites were located between 0.5 and 1 km upstream of newly constructed weirs to ensure that adult chironomids collected at the sampling sites accurately represented the chironomid fauna of the rivers’ lentic habitats.

In June and August 2015, adult chironomids were sampled using a 30 cm diameter sweep net (10 sweeps from grass along a 10 m transect; Figure 2e) for quantitative analysis and using a light trap (Figure 2f) as well as a sweep net for qualitative purposes, and then they were preserved in 80% or 99.5% ethanol.

### 2.2. Morphological Identification

Some chironomid genera are difficult to identify at the species level without slide mounts of specific body parts; therefore, the antennae, head, wings, abdomen, and hypopygium of the adult samples were dissected using a fine needle under a dissecting microscope (SZ61; Olympus, Tokyo, Japan). To prepare permanent slides, the body parts were mounted with Hoyer’s medium and dried for 2–4 days [24]. All specimens were identified using available identification keys and references [5,6,7]. The remaining body parts of each adult specimen were preserved in 99.5% ethanol prior to DNA extraction. All specimens were deposited at the Entomological Museum of Korea University (KU) in Seoul, Korea. The terminology used in this study generally follows that of Sæther [25], Oliver and Dillon [26], and Langton and Pinder [27].

### 2.3. Sequence Generation and Analysis

To verify the correctness of the species identification of the morphologically classified midges and to construct DNA barcodes, total genomic DNA was extracted from the thoracic pre-episternum of each specimen using a DNeasy Blood and Tissue kit (Qiagen, Hilden, Germany). Standard PCR amplification and sequencing protocols were used to generate *COI* fragment sequences. Briefly, the target fragment of *COI* was amplified in 20 μL reactions containing AccuPower PCR PreMix (Bioneer Co., Daejeon, Korea), 1 U Top DNA polymerase, dNTPs (10 mM), Tris-HCl (pH 9.0), KCl (30 mM), MgCl2 (1.5 mM), 1–3 μL (5–50 ng) template DNA, and 1 μL of each primer (LCO1490 and HCO2198; 10 pM each) [28]. Amplification was performed using the following thermal cycling program: 94 °C for 5 min; 35 cycles of 94 °C for 0.5 min; 48 °C for 1 min; 72 °C for 1.5 min; and a final extension at 72 °C for 10 min. The reaction products were separated on 1.5% agarose gels, visualized using UV light, purified, and then sequenced using an ABI Prism 3130 Genetic Analyzer (Applied Biosystems, Foster, CA, USA).

The sequences were aligned using the CLC Main Workbench (version 7.8.1; CLC bio, Aarhus, Denmark), and the results were crosschecked using the ClustalW algorithm in MEGA 7.0 [29]. For phylogenetic relationships between large Korean river chironomid species, a maximum likelihood (ML) tree was constructed using MEGA 7.0, and the GTR + I + Γ model of nucleotide substitution was selected as the best-fitting model for the *COI* sequences using jModelTest (version 2.1.10) based on the Akaike information criterion [30,31]. The convergence diagnostic was calculated for 1000 generations in ML, and groups with a frequency greater than 50% were retained.

Pairwise sequence divergence within and between genetic clusters was calculated using the K2P model in MEGA 7.0. Intraspecific and interspecific genetic distance analyses were performed using the K2P and maximum composite likelihood models of Tamura et al. [32], which were the best-fit nucleotide substitution and base frequency models, respectively, as indicated by MEGA 7.0.

A model-based species depiction analysis showed that species identification did not depict species. However, the phylogenetic tree formed a mono-clade, and the average barcode gap of native chironomids was 3% [33]. In addition, for *Tanytarsus*, which has many unknown and cryptic species, Lin et al. [34] proposed a threshold of 4–5% for the species. Therefore, we identified species that were higher than commonly known thresholds. Therefore, even if model-based species depictions have not been tested, there is little chance of species classification error.

Sequences of *COI* of Korean chironomids from large rivers obtained in the current study were deposited in GenBank under accession numbers OM974370–OM974448.

### 2.4. Chironomid Community Analysis

To evaluate the structure of the chironomid community in the study area, the dominance index (DI), diversity index (H’), richness index (RI), and evenness index (J’), which indicate the quality of the community structure, were calculated as the proportion of the relative density of the primary and secondary species in the community [35] based on the Shannon–Weaver function [36] and by using the formula of Margalef [37] and Pielou [38], respectively.

## 3. Results

### 3.1. Morphology- and DNA-Based Species Identification

A total of 2168 individuals belonging to 40 species, 21 genera, and 3 subfamilies of Chironomidae were collected and identified by quantitative sampling. The identity of most species distinguished morphologically (*n* = 37) was confirmed using *COI* sequence analysis. However, based on morphological and molecular data, one species (Chironominae) could not be assigned as a valid chironomid species. Only three of the remaining species (*Cladopelma edwardsi*, *Corynoneura* sp., and *Einfeldia pagana*) could be identified by morphology alone, owing to a lack of sufficient specimens for DNA analysis. Furthermore, most of the species (*n* = 30) belonged to genera of the subfamily Chironominae, namely, *Benthalia* (one species), *Chironomus* (four species), *Cladopelma* (one species), *Cryptochironomus* (one species), *Dicrotendipes* (three species), *Einfeldia* (one species), *Endochironomus* (one species), *Glyptotendipes* (one species), *Harnischia* (one species), *Lipiniella* (one species), *Microchironomus* (one species), *Parachironomus* (two species), *Phaenopsectra* (one species), *Polypedilum* (seven species), *Tanytarsus* (three species), and an unknown genus (one species). Meanwhile, six species belonged to the genera of the subfamily Orthocladiinae, namely *Corynoneura* (one species), *Cricotopus* (four species), and *Nanocladius* (one species), and four species belonged to the genera of subfamily Tanypodinae, namely *Ablabesmyia* (two species), *Procladius* (one species), and *Tanypus* (one species). Three of the identified species (*Polypedilum kyotoensis*, *P. okiharaki*, and *P. sordens*) were new additions to the documented chironomid fauna of South Korea (Table 1).

### 3.2. DNA Barcode Library

A total of 124 *COI* sequences from 37 species were generated from 81 collected specimens (Han River = 23, Geum River = 26, Yeongsan River = 14, and Nakdong River = 18) and combined with 43 sequences from the GenBank database to establish a DNA barcode dataset (Table 1). An analysis of the sequences confirmed the monophyly of all the sampled subfamilies, except Orthocladiinae, since the analysis indicated that the Orthocladiinae species *Nanocladius tamabicolor* belonged to Tanypodinae. In addition, clustering of the 81 *COI* sequences in the ML tree was congruent with the accepted delineations for most morphospecies. Clades corresponding to 33 (91.7%) of the 36 chironomid species (excluding one invalid Chironominae species) were strongly supported, with a bootstrap value of >95% (Figure 3). Exceptions occurred for several taxa in which the level of deep sequence divergence was observed between individuals assigned to the same morphospecies (*Benthalia carbonaria, Dicrotendipes nervosus*, and *Ablabesmyia longistyla*; Figure 3).

The range of intraspecific divergence values for the studied species was 0–15.0%, with an overall mean of 3.0% (SE = 0.6%). Minimum intraspecific divergence (0%) was observed in *Polypedilum japonicum* and *Tanypus punctipennis*, whereas high levels of intraspecific divergence (>2.3%) were observed in 15 of the 36 species, with the maximum intraspecific divergence (15%) observed in *A. longistyla*. The minimum interspecific divergence (0.5%) was observed between *Cricotopus tricinctus* and *Cricotopus sylvestris*, and the maximum interspecific divergence (28.4%) was observed between *Polypedilum nubifer* and Chironominae species. These include species complexes for which the taxonomic status was unresolved by morphological methods.

### 3.3. Distribution of Chironomids in Large Rivers of South Korea

In this study, 25 species (16 genera in 3 subfamilies) were collected from the Han River; 28 species (17 genera in 3 subfamilies) from the Geum River; 18 species (12 genera in 2 subfamilies) from the Yeongsan River; and 29 species (19 genera in 3 subfamilies) from the Nakdong River. The total relative density was dominated by specimens attributed to the Chironominae (1711 specimens, 79%), which included members of Chironomini (1065 specimens, 62%) and Tanytarsini (646 specimens, 38%), followed by subfamilies of Orthocladiinae (311 specimens, 14%) and Tanypodinae (146 specimens, 7%). Furthermore, the most abundant species of Chironominae was *Tanytarsus tamagotoi* (40 ind./m^2^), whereas the most abundant species of the Orthocladiinae and Tanypodinae were *Nanocladius tamabicolor* (7 ind./m^2^) and *Procladius choreus* (7 ind./m^2^), respectively.

Seven species of Chironominae (*Dicrotendipes nervosus*, *D. pelochloris*, *Glyptotendipes tokunagai*, *Lipiniella moderata*, *Microchironomus tener*, *Polypedilum nubeculosum*, and *T. tamagotoi*), one species of Orthocladiinae (*C. sylvestris*), and two species of Tanypodinae (*P. choreus* and *Tanypus punctipennis*) were found in all four rivers (Table 1).

The mean and standard deviation of the chironomid community indices are listed in Table 2. The DI of the Geum River (0.78 ± 0.20), which accounted for more than 40% of the total population, was the highest among the four rivers, whereas that of the Nakdong River (0.51 ± 0.12) was the lowest. However, H’, which indicates relative community balance and complexity, was the highest in the Nakdong River community (2.70 ± 0.46) and the lowest in the Geum River community (1.76 ± 0.77). Similarly, both RI and J’ were lower in the Geum River community (1.43 ± 0.84 and 0.78 ± 0.16, respectively) than in the other river communities.

The most dominant chironomid species throughout the nine sampling sites was *T. tamagotoi*, while *L. moderata*, *M. tener*, *Tanytarsus formosanus*, and *P. choreus* were locally dominant at some of the sampling sites (Table 2).

## 4. Discussion

Freshwater biologists commonly use chironomid larvae to assess and monitor environmental conditions, particularly the levels of organic pollution in streams, rivers, and wetlands [39,40,41]. Unfortunately, chironomid larvae are usually difficult to identify morphologically, and species identification mostly relies on an analysis of adult males, which tend to possess more species-specific characteristics [1,16]. Accordingly, the DNA barcode library developed in the present study may facilitate the accurate identification of larval chironomids. Moreover, even though *COI* sequences for some Korean chironomids have been fragmented in previous studies [33], the DNA barcode library developed here for chironomids of large South Korean rivers could facilitate a wide variety of biological and environmental projects, such as biomonitoring of large rivers. In addition, 31 species of chironomids were morphologically classified in this study; however, the usefulness of the DNA barcode library was clearly demonstrated by identifying six additional species for a total of 37 species.

In this study, 40 chironomid species, comprising ~13% of the chironomid species distributed in South Korea, were identified using morphological and molecular methods. The composition of predominant chironomid groups in rivers and lowland streams is well known, comprising the subfamilies Chironominae, Orthocladiinae, and Tanypodinae [9,39]. Chironominae was the predominant group in this study, followed by the subfamilies Orthocladiinae and Tanypodinae. However, Podonominae and Diamesinae subfamilies, as well as certain groups of Orthocladiinae, were not included in this study because we focused on midges inhabiting rivers located in urban centers and did not include headwaters or pristine waterways [42].

Among the 40 species collected, the species of the genera *Chironomus* (*C. circumdatus, C. kiiensis, C. nipponensis*, and *C. plumosus*), *Glyptotendipes* (*G. tokunagai*), *Polypedilum* (*P. nubifer*), *Tanytarsus* (*T. formosanus*, *T. oyamai*, and *T. tamagotoi*), *Cricotopus* (*C. sylvestris*), and *Procladius* (*P. choreus*) were collected in large numbers using a light trap, which may indicate that these species are responsible for nuisance activity, such as disturbances in city lights and of outdoor activities. In addition, the genera *Chironomus, Polypedilum*, *Tanytarsus*, and *Cricotopus* occur more frequently in polluted rivers in Asia [39], as observed in this study. Furthermore, *Tanytarsus* is the most abundant species across the four major river sites and has been used as an indicator of moderately polluted streams [43]. In this study, the three *Polypedilum* species (Chironominae) recorded for the first time in Korea were identified by DNA barcoding.

The average intraspecific divergences (0.9–2.32%) were previously reported in Chironomidae [44,45]; however, some researchers have suggested a cryptic diversity range of 2–5% [46,47] or an average intraspecific threshold of 4–8% for members of the genera *Tanytarsus* and *Polypedilum* [34,48]. In this study, more than 8% intraspecific divergence was observed in 3 (*D. nervosus*, *M. tener*, and *A. longistyla*) of the 37 species. Cases of deep intraspecific divergence can reflect misidentifications, cryptic taxa, ancestral polymorphisms, or introgression [49]. No significant morphological differences were found between the species divided into the two groups of *D. nervosus* (Figure 3). Therefore, this species includes cryptic species. *Ablabesmyia longistyla* also exhibited 15% divergence at the divergence level of the Korean species. *Ablabesmyia* includes several other cryptic species, and the final resolution requires a detailed taxonomic study of the entire group. By contrast, *C. sylvestris* and *C. tricinctus* were grouped together (Figure 3). Considering that the average genetic distance between *C. sylvestris* and *C. tricinctus* was 0.5% (see Appendix A), these specimens could be the same species or two closely related sibling species. From a taxonomic perspective, *Cricotopus* is one of the most difficult genus in Chironomidae to identify because of its lack of reliable diagnostic characters [45]. In general, *COI* barcodes for each species formed a distinct cluster separated from its nearest neighbor, but there were exceptions. Some of these cases involved unusually large intraspecific distances, whereas in other cases, there was little or no separation between species. Where barcodes failed to distinguish between species, the taxa involved were generally morphologically similar and closely related [49]. Therefore, species with a limit of *COI* barcode analysis such as those of *Cricotopus* require specialist entomologists’ support and the collection of as many representatives of each type as possible to obtain reasonable estimates of intraspecific variation [50,51].

## 5. Conclusions

A total of 124 *COI* sequences were established from 37 species belonging to 19 genera, and the resulting DNA barcode library effectively discriminated >90% of riverine Chironomidae in South Korea. Ten species, which are considered indicator species for large rivers, were collected from all four large rivers, and members of Chironominae occurred more frequently than members of other subfamilies, with *Tanytarsus tamagotoi* being the most common and widespread species in large rivers in South Korea. This study contributes to the current knowledge of riverine chironomids, which have the potential to cause environmental and public health problems for residents of large cities in Asia, including South Korea.

## Figures and Tables

**Figure 1 insects-13-00346-f001:**
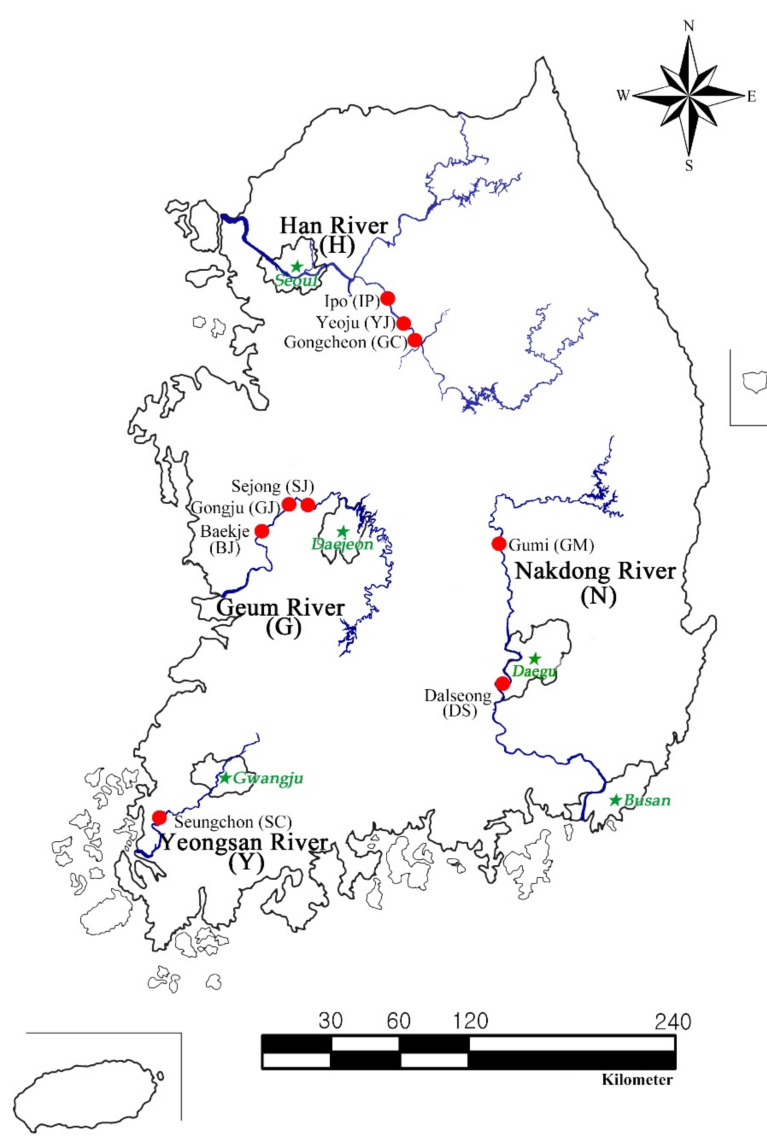
Chironomid sampling sites in the four large rivers in South Korea.

**Figure 2 insects-13-00346-f002:**
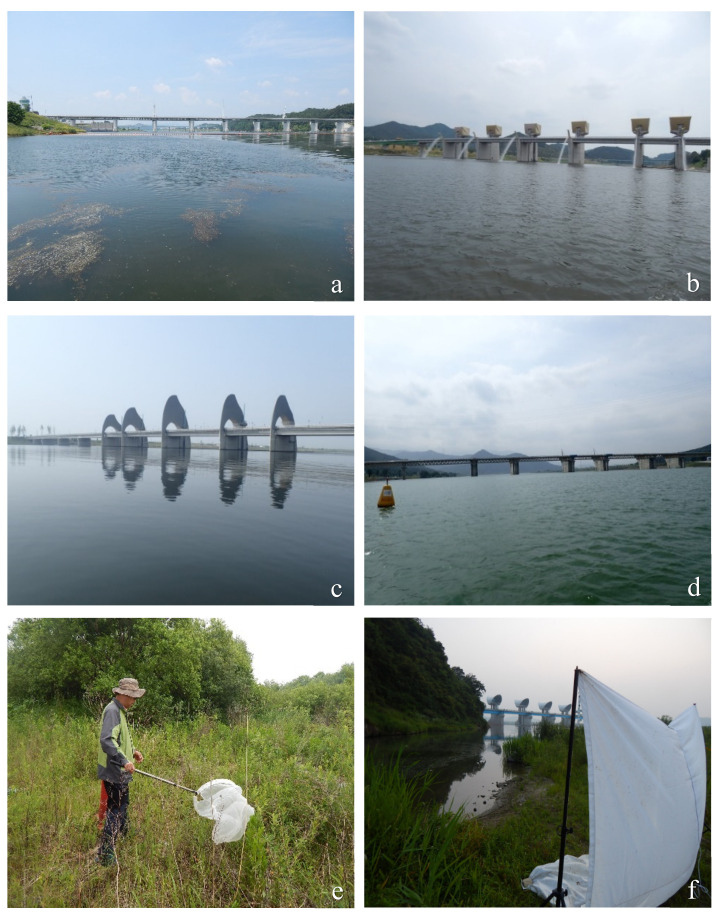
Chironomid sampling sites and collection methods. (**a**) Han River at Ipo weir; (**b**) Geum River at Sejong weir; (**c**) Yeongsan River at Seungchon weir; (**d**) Nakdong River at Danlseong weir; (**e**) sweeping along Han River at Ipo weir; (**f**) light-tarping along Yeongsan River at Seungchon weir.

**Figure 3 insects-13-00346-f003:**
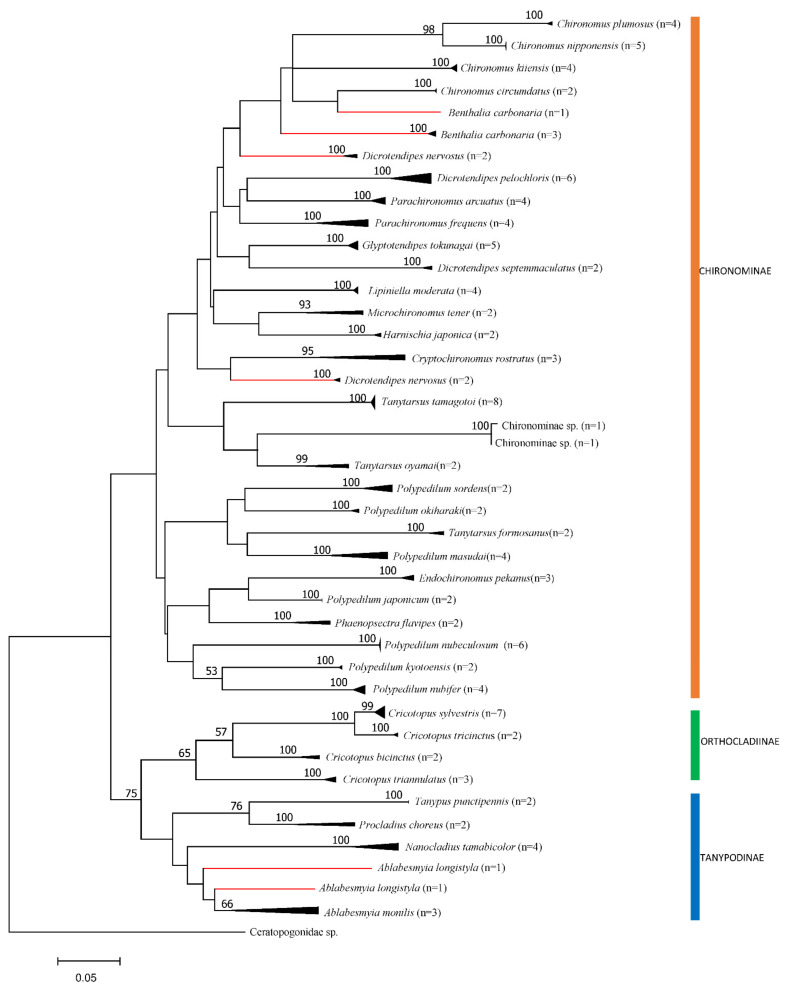
Maximum likelihood tree of *COI* sequences from chironomids collected from large rivers in South Korea. Node values indicate bootstrap values (1000 replicates) of >50%. Scale bar corresponds to 0.05 changes per nucleotide.

**Table 1 insects-13-00346-t001:** Distribution of Korean Chironomidae species, and the mean of intraspecific divergence of 37 species.

Sub -Families	Species	Identification Methods	Rivers				No. of Seq.	Intraspecific Divergence (%)	GenBank Accession No.
			H	G	Y	N			
Chironominae	*Benthalia carbonaria* (Meigen, 1804)	M/D	+	+		+	4(2)	7.0	^†^ AB838654, ^†^ JF412115, OM974370, OM974371
	*Chironomus circumdatus* (Kieffer, 1916)	M/D		+	+		2(1)	0.2	^†^ KY845435, OM974383
	*Chironomus kiiensis* Tokunaga, 1936	M/D		+	+	+	4(1)	0.4	^†^ JF412086, OM974380, OM974381, OM974382
	*Chironomus nipponensis* Tokunaga, 1940	M/D	+	+		+	5(1)	0.2	^†^ JN887050, OM974375, OM974376, OM974377, OM974378
	*Chironomus plumosus* (Linnaeus, 1758)	M/D		+		+	2(1)	0.8	^†^ JF412098, OM974379
	*Cladopelma edwardsi* (Kruseman, 1933)	M		+		+	-	-	
	*Cryptochironomus rostratus* Kieffer, 1921	M/D		+	+	+	3(1)	7.3	^†^ KP902749, OM974384, OM974385
	*Dicrotendipes nervosus* (Staeger, 1839)	M/D	+	+	+	+	4(1)	9.9	^†^ JF412131, OM974386, OM974387, OM974388
	*Dicrotendipes pelochloris* (Kieffer, 1912)	M/D	+	+	+	+	6(1)	3.7	^†^ JF412111, OM974390, OM974391, OM974392, OM974393, OM974394
	*Dicrotendipes septemmaculatus* (Becker, 1908)	M/D	+	+			2(1)	1.5	^†^ HQ846345, OM974389
	*Einfeldia pagana* (Meigen, 1838)	M				+	-	-	
	*Endochironomus pekanus* Kieffer, 1916	M/D			+		3(2)	1.7	^†^ KP902763, ^†^ AB838660, OM974395
	*Glyptotendipes tokunagai* Sasa, 1979	M/D	+	+	+	+	5(1)	1.3	^†^ JF412126, OM974396, OM974397, OM974398, OM974399
	*Harnischia japonica* Hashimoto, 1984	M/D		+		+	2(1)	1.0	^†^ LC329112, OM974400
	*Lipiniella moderata* Kalugina, 1970	D	+	+	+	+	4(1)	0.3	^†^ JF412078, OM974372, OM974373, OM974374
	*Microchironomus tener* Kieffer, 1916	M/D	+	+	+	+	2(1)	8.9	^†^ KJ188143, OM974401
	*Parachironomus arcuatus* (Goetghebuer, 1919)	M/D	+	+		+	4(2)	1.6	^†^ JF412133,^†^KP902786, OM974402, OM974403
	*Parachironomus frequens* (Johannsen, 1905)	M/D	+	+		+	4(1)	3.9	^†^ KM571019, OM974404, OM974405
	*Phaenopsectra flavipes* (Meigen, 1818)	D	+	+			2(1)	5.2	^†^ KC250831, OM974406
	*Polypedilum japonicum* (Tokunaga, 1938)	M/D		+		+	2(1)	0.0	^†^ LC329191, OM974407
	*Polypedilum kyotoensis ** (Tokunaga, 1938)	D		+	+		2(1)	0.4	^†^ MG950080, OM974408
	*Polypedilum masudai* (Tokunaga, 1938)	M/D		+	+		4(2)	6.3	^†^ KU497070,^†^MG949781, OM974409, OM974410
	*Polypedilum nubifer* (Skuze, 1889)	M/D			+	+	5(2)	0.8	^†^ MG950024,^†^LC329202, OM974411, OM974412, OM974413
	*Polypedilum okiharaki* * Sasa, 1990	D	+				2(1)	1.1	^†^ MG949962, OM974414
	*Polypedilum sordens* * (Wulp, 1874)	D	+	+			4(2)	2.7	^†^ MG950038, ^†^ MG949841, OM974415, OM974416
	*Polypedilum nubeculosum* (Meigen, 1804)	M/D	+	+	+	+	6(1)	0.4	^†^ JF412161, OM974417, OM974418, OM974419, OM974420, OM974421
	*Tanytarsus formosanus* Kieffer, 1912	D				+	2(1)	2.4	^†^ KT613431, OM974422
	*Tanytarsus oyamai* Sasa, 1979	M/D			+	+	2(1)	6.4	^†^ LC329286, OM974423
	*Tanytarsus tamagotoi* Sasa, 1983	M/D	+	+	+	+	8(1)	0.3	^†^ JF412175, OM974424, OM974425, OM974426, OM974427, OM974428, OM974429, OM974430
	*Chironominae* sp.	U	+			+	2(0)	0.5	OM974431, OM974432
Orthocladiinae	*Corynoneura* sp.	M	+				-	-	
	*Cricotopus bicintus* (Meigen, 1818)	M/D	+			+	2(1)	3.5	^†^ JN887058, OM974433
	*Cricotopus sylvestris* (Fabricius, 1794)	M/D	+	+	+	+	7(2)	1.0	^†^ JN887068,^†^LC329073, OM974434, OM974435, OM974436, OM974437, OM974438
	*Cricotopus triannulatus* (Macquart, 1826)	M/D	+				3(1)	1.6	^†^ LC050962, OM974439, OM974440
	*Cricotopus tricinctus* (Meigen, 1818)	M/D	+	+		+	2(1)	0.5	^†^ AB838617, OM974441
	*Nanocladius tamabicolor* Sasa, 1981	D				+	4(1)	5.7	^†^ LC050919, OM974442, OM974443, OM974444
Tanypodinae	*Ablabesmyia longistyla* Fittkau, 1962	D	+	+		+	2(1)	15.0	^†^ JN887044, OM974445
	*Ablabesmyia monilis* (Linnaeus, 1758)	M/D	+				3(1)	1.2	^†^ JN887045, OM974446
	*Procladius choreus* (Meigen, 1804)	M/D	+	+	+	+	2(1)	7.4	^†^ JN887097, OM974447
	*Tanypus punctipennis* Meigen, 1818	M/D	+	+	+	+	2(1)	0.0	^†^ JN887099, OM974448
No. of species found in each river			25	28	18	29			

M (morphology), D (DNA barcode), U (unknown), + (species found in the indicated river), and N (the number of species; the reference sequences are in parentheses). The names of the rivers correspond to the abbreviations used in Figure 1. Asterisks indicate species recorded as new to the Korean fauna. GenBank accession numbers with daggers (^†^) are specimens with sequences acquired from GenBank.

**Table 2 insects-13-00346-t002:** Summary of species richness and biotic indices using quantitative data of Chironomidae in the four major rivers in South Korea.

	Site	GPS	No. of			DI	H’	RI	J’	Dominant Species
			Genus	Species	Individuals					
Han River	IP	37°24′7.9″, 127°32′23.46″	8	11	216	0.56 ± 0.06	2.56 ± 0.02	2.09 ± 0.26	0.77 ± 0.12	*Tanytarsus tamagotoi*
	YJ	37°19′30.92″, 127°36′39.72″	11	13	132	0.50 ± 0.06	2.92 ± 0.21	2.52 ± 0.24	0.83 ± 0.00	*Tanytarsus tamagotoi*
	GC	37°16′33.96″, 127°41′4.58″	10	14	139	0.79 ± 0.01	2.50 ± 0.23	2.06 ± 0.39	0.79 ± 0.06	*Microchironomus tener*
Total			14	19	504	0.55 ± 0.06	2.66 ± 0.25	2.23 ± 0.33	0.80 ± 0.07	
Geum River	SJ	36°28′25.11″, 127°15′46.06″	11	11	78	0.83 ± 0.24	1.72 ± 1.08	1.42 ± 1.32	0.84 ± 0.17	*Procladius choreus*
	GJ	36°27′51.21″, 127°05′59.65″	11	13	66	0.60 ± 0.12	2.41 ± 0.20	1.98 ± 0.26	0.83 ± 0.03	*Lipiniella moderata*
	BJ	36°19′16.6″, 126°56′34.03″	8	8	160	0.92 ± 0.12	1.16 ± 0.46	0.89 ± 0.73	0.67 ± 0.24	*Tanytarsus tamagotoi*
Total			16	20	304	0.78 ± 0.20	1.76 ± 0.77	1.43 ± 0.84	0.78 ± 0.16	
Yeongsan river	SC	35°03′55.2″, 126°45′59.5″	9	10	22	0.55 ± 0.03	2.39 ± 0.22	2.14 ± 0.11	0.94 ± 0.04	*Tanytarsus tamagotoi*
Total			9	10	22	0.55 ± 0.03	2.39 ± 0.22	2.14 ± 0.11	0.94 ± 0.04	
Nakdong river	GM	36°14′11.3″, 128°20′43.6″	11	14	75	0.53 ± 0.19	2.60 ± 0.76	2.10 ± 0.40	0.90 ± 0.01	*Tanytarsus tamagotoi*
	DS	35°44′04.7″, 128°25′02.1″	11	14	108	0.49 ± 0.03	2.80 ± 0.12	2.07 ± 0.60	0.89 ± 0.03	*Tanytarsus formosanus*
Total			17	23	183	0.51 ± 0.12	2.70 ± 0.46	2.09 ± 0.42	0.89 ± 0.02	

See locality abbreviations in Figure 1.

## Data Availability

Data are available upon request from the authors.

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
