# Peer review of "Diversity and DNA Barcode Analysis of Chironomids (Diptera: Chironomidae) from Large Rivers in South Korea"

_insects, 2022, doi:10.3390/insects13040346_

Round 1
Reviewer 1 Report
Dear authors
your paper entitled "Diversity and DNA barcode analysis of chironomids (Diptera: Chironomidae) from large rivers in South Korea" is interesting but it needs improvements before publication. The major problem is that you did not explain in detail what you did. Species delimitation analysis is very important and should be carried out carefully to avoid the identification of "false" species. I didn't find anything about this analysis in your paper and I think that you should explain how you identified the different species and you should carry out the species delimitation analysis.
Details of my comments are in the attached file

Author Response
-
Thank you very much for your insightful feedback to our paper. We have revised our manuscript based on your comments and made point-by-point responses, below.
1. General comments:
Your paper entitled "Diversity and DNA barcode analysis of chironomids (Diptera: Chironomidae) from large rivers in South Korea" is interesting but it needs improvements before publication. The major problem is that you did not explain in detail what you did. Species delimitation analysis is very important and should be carried out carefully to avoid the identification of "false" species. I didn't find anything about this analysis in your paper and I think that you should explain how you identified the different species and you should carry out the species delimitation analysis.
Response: Thank you for this important comments. We basically identified our adult chironomid species based on traditional morphological characters such as male genitalia and then carefully accessed species delimitation using COI barcode data. According to previous studies, a threshold of 0.9–5% was suggested for some groups of Chironomidae (Ekrem et al, 2007; Lin et al, 2015, Silva & Wiedenbrug, 2014). Kim et al. (2012) also showed a 3% barcode gap for some groups of Korean Chironomidae.
We add the following sentences (Page 5 , lines 141–147):
“Species identification did not depict species using model-based species depiction analysis. However, the phylogenetic tree forms a mono-clade, and the average barcode gap of native chironomids is 3% by Kim et al (2012). Also, for the genus Tanytarsus, which has many unknown and cryptic species, Lin et al (2015) proposed a threshold of 4-5% for the species. Therefore, we identified species that were higher than the commonly known threshold. Therefore, even if model-based species depictions have not been tested, there is little chance of a species classification error..”
- 2. Pages 6–7, Table 1.
It is not clear to me the meaning of these numbers. You should explain better them. Are they the number of species found in each river? and what is the meaning of “O”? Perhaps, presence of that species in that river?
Response: This number means the “No. of species found in each river” and is written in Table 1 (page 7) for clarity. We changed the symbol “O” to “+” to avoid confusion meaning “presence” that explained in the table legend.
- Page 7, Line 177
In table 1, 40 species are reported. They are more than the 37 you identified and less than those included in the barcode dataset. I think you should explain better what you did.
Response: refer to page 5, lines 161–166:
“The identity of most morphological identified species (n = 37) was confirmed by COI sequence analysis. However, based on morphological and molecular data, one species (Chironominae sp.) could not be assigned to a valid chironomid species. Although, three of the remaining species (Cladopelma edwardsi, Corynoneura sp., and Einfeldia pagana) could only be identified morphologically, due to a lack of sufficient specimens for DNA analysis.”
How did you identify the different species? Did you use a genetic divergence threshold? Did you use the BIN? How did you carry out the species delimitation analysis?
Response: Please, see general comments (#1), above.
- Page 7, Line 195
"The taxonomic status is unresolved by morphological methods" Why?
Response: Chironominae sp. in our study showed a distinct morphological difference from other recorded species. It is very possible that this is a new species, but barcode information is not available in the public database (NCBI). However, we deleted this sentence to avoid a confusion.
- Page 9, Line 227
I understood that you analyzed the COI DNA barcode to identify the Chironomidae species found in the four rivers under study. You identified 37 species in this way. But how did you identify all the species found in the four rivers? Again, I think that you need to improve your paper explaining in more detail your study.
Response: As explained above, we basically identified the species by morphological taxonomic characters such as male genitalia. Then, the 37 species were verified using DNA barcodes excluding the 3 species that DNA was not available.
(See general comments #1, above)
The revised manuscript is attached.
Thanks.

Reviewer 2 Report
The authors have made a valuable contribution to science by producing a well written paper documenting their comprehensive study of the distribution chironomids associated with four major rivers in Korea using genetics and morphology. The DNA barcode library they produced using CO1 will be a useful tool for stream ecologists undertaking bioassessment of other freshwater habitats. As the authors state, the identification of chironomid species, particularly those from the subfamilies Chironominae and Orthocladiinae can be a challenge using microscopy alone; this DNA library will help improve the accuracy of identification.
The figures and tables included are effective and informative.
Review comments/suggestions
Simple summary
Page 1, Line 15
…..distributions and community structures – use singular rather than plural here
Abstract
Page 1, Line 21
Outbreaks of chironomids constitute more of a nuisance than a “public health risk” as they aren’t known to be disease vectors. I suggest removing the reference to public health.
Page 1, Lines 24-25
distributions and community structures
Introduction
Page 1, Lines 44-45
on of larval or pupal stages are……..
Page 3, Line 67
…….causing a lot of damage to people…..
I suggest removing this reference to chironomids being a serious risk to public health. Detecting an elevated immune response to a few chironomid species in a relatively small percentage of individuals tested isn’t a serious medical issue.
Results
Page 6, Table 1.
I suggest aligning all text and numeric data to the left (as you have in Table 2) rather than using centred alignment which looks a little messy. The table would be more easily interpreted if the legend at the base of the table on page 7 was moved to the caption on lines 167 and 168.
Page 7, Lines 170-171 Table Legend
The names of rivers corresponded with to the abbreviations used in Figure 1. Asterisks indicate d species recorded as being new newly recoded species to the Korean fauna.
Discussion
Page 11, Line 244
……DNA barcode library was well clearly demonstrated to by identifying identified 37 species through using the DNA alone.
Page 11, Line 248
…….groups of in rivers and lowland streams is well known, comprising subfamilies Chironominae, Orthocladiinae, and Tanypodinae….
Page 11, Lines 251 - 253
“However, the Podonominae and Diamesinae subfamilies, as well as certain groups of Orthocladiinae are not included in this study as they are more diverse and frequent in higher-order streams, mountain rivers, and forest areas”.
This sentence seems contradictory because orthoclad chironomids have been included in the study. You stated the scope of the study from the outset – headwaters and pristine waterways haven’t been included.
Page 11, Line 272
…..between the two paraphylotic paraphyletic groups of……
Page 11, Line 277
……C. tricinctus was 0.5% in on average………
Page 11, Line 281
……and those species mentioned above……….
Author Response
Thank you very much for your insightful feedback to our paper. We have revised our manuscript based on your comments and made point-by-point responses, below.
Reviewer: 2
- Page 1, Line 15
…..distributions and community structures – use singular rather than plural here
Response: Corrected.
- Page 1, Line 21
Outbreaks of chironomids constitute more of a nuisance than a “public health risk” as they aren’t known to be disease vectors. I suggest removing the reference to public health.
Response: Agree. We have deleted the word “public health”.
- Page 3, Line 67
…….causing a lot of damage to people…..
I suggest removing this reference to chironomids being a serious risk to public health. Detecting an elevated immune response to a few chironomid species in a relatively small percentage of individuals tested isn’t a serious medical issue.
Response: We agree that “causing a nuisance to people” would be more accurate and precise. Changed (see page 3, line 67)
- Page 6, Table 1.
I suggest aligning all text and numeric data to the left (as you have in Table 2) rather than using centred alignment which looks a little messy. The table would be more easily interpreted if the legend at the base of the table on page 7 was moved to the caption on lines 167 and 168.
Response: We aligned some text and numeric data to the left side.
- Page 7, Lines 170–171 Table Legend
The names of rivers corresponded with to the abbreviations used in Figure 1. Asterisks indicated species recorded as being new newly recoded species to the Korean fauna.
Response: We edited your comments by page 7, lines 182–184.
- Line 244: ……DNA barcode library was well clearly demonstrated to by identifying identified 37 species through using the DNA alone.
Line 248: …….groups of in rivers and lowland streams is well known, comprising subfamilies Chironominae, Orthocladiinae, and Tanypodinae….
Line 272: …..between the two paraphylotic paraphyletic groups of……
Line 277: ……C. tricinctus was 0.5% inon average………
Line 281: ……and those species mentioned above……….
Response: We have reflected this comment by page 12, lines 256, 260–261, 284, 289, and page 13 line 293.
- Page 11, Lines 251-253
“However, the Podonominae and Diamesinae subfamilies, as well as certain groups of Orthocladiinae are not included in this study as they are more diverse and frequent in higher-order streams, mountain rivers, and forest areas”.
This sentence seems contradictory because orthoclad chironomids have been included in the study. You stated the scope of the study from the outset – headwaters and pristine waterways haven’t been included.
Response: We modified the sentence as below:
“However, the Podonominae and Diamesinae subfamilies, as well as certain groups of Orthocladiinae are not included in this study as we focused on midges inhabiting rivers located in urban centers and have not been include headwaters and pristine waterways.” (see page 12, lines 263–266)
The revised manuscript is attached.
Thanks.

Reviewer 3 Report
This is a well-designed and very interesting study that will have general interest beyond the chironomid community. Especially, I like how the design of the study are put in a context of monitoring environmental degradation and urban life in South Korea. As I am not an expert on Chironomidae I cannot judge the validity of the morphological identifications in this study but the thorough reference to methods and literature used for morphological identification is exemplary. The genetic methods and results also seem to be thorough and sound. I am a bit more skeptical to the use of sweep net transects to obtain abundance data based apparently only on two collecting dates each as these inevitably represent a snapshot of the species swarming intensity just on these dates.
I would strongly recommend adding author names and year of publication to all the scientific names presented in Table 1, and also at their first appearance in the text.
I find the section "3.2. DNA barcode library" the only slightly problematic part of the manuscript since a specific dataset that supports the conclusions of this section in a public repository is lacking, despite constructing a (hopefully public) DNA barcode library being heavily emphasized in the abstract. Such a dataset would greatly increase the value of this study, especially in the context discussed in the first section of the discussion. The GenBank accession numbers are missing in Table 1 and it is an essential minimum for the study’s reproducibility that these are added before publication. The supplementary material table neither has any reference to public sequences. There is further no attempt to compare the COI sequences in a wider context, specifically no reference to the largest DNA barcode library in the world, the Barcode of Life online database BoldSystems (BOLD). The species limits are, however, adequately discussed in relation to previous publications in the discussion, but with the current reference library at hand this could be investigated even further. Since I have no access to the actual sequences, I could only check the reference library on BOLD through testing some of the scientific names. By a random check I found that the species Benthalia dissidens and Ablabesmyia longistylahas both have 5 different Barcodes Index Numbers (BINs) each on BOLD, this indicating that these species names may be interpreted too wide or/and differently in different parts of the world. Hence, I would like to have seen your results on these species discussed in a wider context, i.e. how the deep divergence found in these species in South Korea is reflected in COI sequences from other parts of the world. As a minimum, I would like to see your result presented as a concrete dataset alongside this publication to document your interpretation (including those downloaded from GenBank) of the included species in South Korea. Even if only uploaded to GenBank the sequences will soon also be harvested into BOLD and can be interpreted more widely there. Preferably, I would like to see your dataset being uploaded directly to BOLD and presented there as a public chironomid reference library from South Korea.
The comments made on higher classification based on the COI data is ok, but it is important not to put too much phylogenetic emphasis on COI sequences alone, but rater use them to check and define species limits.
The boldface indication of year of publication in the reference list is not implemented for all the references (in the start).
Author Response
Thank you very much for your insightful feedback to our paper. We have revised our manuscript based on your comments and made point-by-point responses, below.
Reviewer: 3
- I would strongly recommend adding author names and year of publication to all the scientific names presented in Table 1, and also at their first appearance in the text.
Response: Information on species is added in Table 1.
- I find the section "3.2. DNA barcode library" the only slightly problematic part of the manuscript since a specific dataset that supports the conclusions of this section in a public repository is lacking, despite constructing a (hopefully public) DNA barcode library being heavily emphasized in the abstract. ‧‧‧‧‧ Even if only uploaded to GenBank the sequences will soon also be harvested into BOLD and can be interpreted more widely there. Preferably, I would like to see your dataset being uploaded directly to BOLD and presented there as a public chironomid reference library from South Korea.
Response: We added a sentence:
“Sequences of COI of Korean chironomid in large rivers obtained in the current study were deposited in GenBank under accession numbers OM974370–OM974448, respectively. ”(see page 5, lines 148–149)
- The boldface indication of year of publication in the reference list is not implemented for all the references (in the start).
Response: Corrected.
The revised manuscript is attached.
Thanks.

Round 2
Reviewer 1 Report
Dear authors
I think that the paper has been improved by adding the requested details, and it can be published as is it.
Author Response
Thanks for your useful review and kind comments. I agree with your opinions. I have improved discussion on page 12 (lines 294-298) and page 13 (lines 299-301) including suggested literature of Montagna 2016 (#50). Thanks for this useful information.
Additional minor corrections were done throughout the manuscript.
English language was double checked by a professional native English editor.
Thanks again for your time and efforts.
